# Extracellular Signalling Modulates Scar/WAVE Complex Activity through Abi Phosphorylation

**DOI:** 10.3390/cells10123485

**Published:** 2021-12-10

**Authors:** Shashi Prakash Singh, Peter A. Thomason, Robert H. Insall

**Affiliations:** 1CRUK Beatson Institute, Glasgow G61 1BD, UK; S.Singh@beatson.gla.ac.uk (S.P.S.); p.thomason@beatson.gla.ac.uk (P.A.T.); 2Institute of Cancer Sciences, University of Glasgow, Glasgow G61 1QH, UK

**Keywords:** pseudopodia, phosphorylation, F-actin, Scar/WAVE complex, migration

## Abstract

The lamellipodia and pseudopodia of migrating cells are produced and maintained by the Scar/WAVE complex. Thus, actin-based cell migration is largely controlled through regulation of Scar/WAVE. Here, we report that the Abi subunit—but not Scar—is phosphorylated in response to extracellular signalling in *Dictyostelium* cells. Like Scar, Abi is phosphorylated after the complex has been activated, implying that Abi phosphorylation modulates pseudopodia, rather than causing new ones to be made. Consistent with this, Scar complex mutants that cannot bind Rac are also not phosphorylated. Several environmental cues also affect Abi phosphorylation—cell-substrate adhesion promotes it and increased extracellular osmolarity diminishes it. Both unphosphorylatable and phosphomimetic Abi efficiently rescue the chemotaxis of Abi KO cells and pseudopodia formation, confirming that Abi phosphorylation is not required for activation or inactivation of the Scar/WAVE complex. However, pseudopodia and Scar patches in the cells with unphosphorylatable Abi protrude for longer, altering pseudopod dynamics and cell speed. *Dictyostelium*, in which Scar and Abi are both unphosphorylatable, can still form pseudopods, but migrate substantially faster. We conclude that extracellular signals and environmental responses modulate cell migration by tuning the behaviour of the Scar/WAVE complex after it has been activated.

## 1. Introduction

In migrating cells, the Scar/WAVE complex acts as a principal catalyst to generate actin-rich protrusions. In particular, lamellipods or pseudopods are generated by activation of the Arp2/3 complex controlled by the Scar/WAVE complex, leading to polymerization and growth of actin filaments [1]. The five-membered complex consists of Pir121, Nap1, Scar/WAVE, Abi and HSPC300 subunits [2]. It is regulated by the small GTPase Rac1. Rac1 does not interact with the complex when GDP-bound. However, when GTP-bound it interacts with the Pir121 subunit, making the Scar/WAVE complex active and able to promote actin polymerization [3,4]. Multiple other regulators affect the Scar/WAVE complex’s activity [1].

Several authors have reported that phosphorylation, of either the Scar/WAVE [5,6,7,8,9,10] and Abi [7,11,12] subunits, is a necessary and important activator of the complex, and is important for the formation of proper lamellipodia or pseudopodia. However, we recently showed that Scar/WAVE phosphorylation is not required for the Scar/WAVE complex to be activated in either mammalian or *Dictyostelium* cells; rather, it modulates pseudopod dynamics [13]. In this work, we investigate if the same is true for Abi.

Several authors have reported phosphorylation of Abi [6,7,11,12,14,15,16]. Phosphorylation by the tyrosine kinase Abl (Abelson kinase) has been shown to control the stability of Abi, as well actin morphology, in both blood cells [11,14] and in *Drosophila melanogaster* [12]. Other reports describe ERK2 phosphorylation sites in Abi [7,14] and suggest that they activate protrusion and control its dynamics during migration, though we note that our recent data strongly oppose the previously described role of ERK2 in Scar/WAVE activation. CDK1-mediated serine phosphorylation in Abi may be important in mitosis [15].

ERK2 and Abl have specific recognition sequences that are not conserved in Abi proteins from different eukaryotic organisms. Moreover, receptor tyrosine kinases are not present in *Dictyostelium* [17,18,19], despite strong conservation in the sequence and activation mechanisms of its Scar/WAVE complex. We have, therefore, investigated the importance of Abi phosphorylation in *Dictyostelium* cell motility.

## 2. Materials and Methods

### 2.1. Dictyostelium Cell Culture

Wild type strains of *Dictyostelium discoideum*, Ax3 and NC4, were obtained from the *Dictyostelium* Stock Center (http://dictybase.org/StockCenter; 5 June 2021) [20]. All cell lines used and generated in this study are mentioned in Appendix A. The axenic culture of cells was maintained in HL5 medium (ForMedium, Norfolk, UK) supplemented with 100 units of penicillin and 100 mg/mL streptomycin-sulfate in Petri dishes or in shaken suspension at 2–4 × 10^6^ cells/mL (22 °C, 125 rpm). Non-axenic culture of cells was maintained on SM agar (ForMedium) plates with *Klebsiella aerogenes*. Cells were mixed in *Klebsiella*
*aerogenes*, spread on SM agar plates with a sterile L-rod spreader and kept in a 22 °C incubator for approximately 40 h to form a clearing zone enriched with amoebal cells.

### 2.2. Plasmid Constructs

A shuttle vector containing the *abi* gene was constructed by excising *abi* from vector pAD48 [21] with BglII-SpeI and cloning it into pDM344. To replace the actin15 promoter from the resulting vector, the 5′UTR of *abi* was PCR-amplified from *Dictyostelium discoideum* genomic DNA and cloned using EcoRI-ClaI. The NgoMIV fragment from the above construct was subcloned into pDM304 and pDM459 to generate Abi and HSPC300-eGFP co-expression constructs. Phosphorylation mutants of Abi were generated by PCR-based site-directed mutagenesis using primers listed in Appendix A. To construct Pir121-EGFP knock-in constructs, the 3′ UTR of Pir121 was PCR amplified from genomic DNA using the primers listed in Appendix A and cloned into pPT808 [4] using Gibson assembly (NEB). All plasmids used in this study are listed in Appendix A.

### 2.3. Generation of Scar-/Abi- Double Knock out Cells

The *scar*-/*abi*- double knockout was generated by knocking out *abi* in *scar*-cells (strain IR46) by homologous recombination using knock out vector pAY16P [22]. Briefly, Scar- cells were transfected with 10 µg KpnI-MluI linearized pAY16P. At 24 h after transfection, *scar*-/*abi*-knockout clones were selected with 10 µg/mL blasticidin in 96 well plates. Knockout clones were confirmed by PCR using *abi* UTR and blasticidin primers mentioned in Appendix A, and Western blotting.

### 2.4. Generation of Scar-/Abi-/pir121-eGFP Cells

The Pir121-eGFP knock in construct was generated by transfecting *scar*-/*abi*-cells with a gel-purified *pir121eGFP* region, excised with EcoRI. eGFP-labelled cells were FACS-sorted 48 h post-transfection. Positive knock in clones were confirmed by Western blotting and AiryScan confocal microscopy.

### 2.5. Transfection of Dictyostelium Cells

To transfect using extrachromosomal plasmids, 1 × 10^7^ cells/mL were washed once with electroporation buffer and resuspended in 420 µL EB (EB; 5 mM Na_2_HPO_4_, 5 mM KH_2_PO_4_ and 50 mM sucrose), and approximately 0.5 μg plasmid DNA was mixed with cells in 2 mm gap electroporation cuvettes and electroporated into the cells by pulsing once at 500 V using an ECM399 electroporator (BTX, Harvard). Cells were then transferred into HL5 medium. After 24 h, 10 μg/mL G418, blasticidin or 50 μg/mL hygromycin were added to select transformants. To make knockout or knockin clones, 10 μg linearized vectors were electroporated into cells.

### 2.6. GFP-TRAP Pulldown

Cells grown in a 15 cm dish were lysed with 1 mL TNE/T buffer (10 mM Tris-HCl pH 7.5, 150 mM NaCl, 0.5 mM EDTA and 0.1% Triton X-100) containing HALT protease and phosphatase inhibitors. Lysates were kept on ice for 5 min and cleared by centrifugation (13,000 rpm, 4 °C, 5 min). A total of 25 μL of GFP-TRAP beads (ChromoTek, Planegg, Germany) were washed twice with TNE/T buffer and resuspended in 100 μL TNE/T buffer, and 1 mg of lysate was added to the beads and kept on rotation for 30 min at 4 °C. Beads were spun down at (2700 g, 4 °C, 2 min) and washed 3 times with TNE/T buffer. Proteins from the beads were eluted by adding 50 μL 2x NuPAGE LDS sample buffer and boiled (100 °C, 5 min).

### 2.7. Western Blotting

Cells were lysed by directly adding NuPAGE LDS sample buffer (Invitrogen, Waltham, MA, USA) containing 20 mM DTT, HALT protease and phosphatase inhibitors (Thermo Fisher Scientific, Waltham, MA, USA) on top of cells and boiled at 100 °C for 5 min. Proteins were separated on 10% Bis-Tris NuPAGE gels (Invitrogen, Waltham, MA, USA) or on hand-poured low-bis acrylamide (0.06% bis acrylamide and 10% acrylamide) gels, then separated at 150 V for 90 min. Proteins were transferred onto 0.45 μM nitrocellulose membrane. Membranes were blocked in TBS + 5% non-fat milk. Primary antibodies were used at 1:1000 dilution, and 1:10000 fluorescently conjugated secondary antibody was used to detect the protein bands by Odyssey CLx Imaging System (LI-COR Biosciences, Lincoln, NE, USA). Mccc1 was used as a loading control [23].

### 2.8. Phosphatase Treatment

Cells grown in a 35 mm Petri dish were lysed in 100 μL TN/T buffer (10 mM Tris-HCl pH 7.5, 150 mM NaCl and 0.1% Triton X-100), kept on ice for 5 min and cleared by centrifugation (13,000 rpm, 4 °C, 5 min). Proteins were dephosphorylated using 1 μL Lambda phosphatase at 30 °C (NEB; P0753S) for 1 h. Protein dephosphorylation was assessed by Western blotting using low-bis gels.

### 2.9. Under Agarose Chemotaxis

Cellular morphology, pseudopod dynamics and cell migration were measured by an under-agarose folate chemotaxis assay, as described earlier [13]. In brief, 0.4% SeamKem GTG agarose was dissolved in boiling Lo-Flo medium (ForMedium). After cooling, 10 μM folic acid was added, and 5 mL of the agarose-folate mix was poured into the 1% BSA-coated 50 mm glass-bottom dishes (MatTek Life Sciences, Ashland, MA, USA). A 5 mm wide trough was cut with a sterile scalpel and filled with 200 μL of 2 × 10^6^ cells/mL. Cell migration was imaged after 4–6 h with 10x and 60x DIC. To examine the localization of labelled proteins in the pseudopods, cells were also imaged using an AiryScan confocal microscope (Zeiss, Jena, Germany).

The migratory parameters, such as the speed of migrating cells, their directedness and chemotaxis efficiency index (CEI), were quantified using a home-made ImageJ plugin. Directedness is the distance between the beginning and the end divided by the total distance travelled. If the cell moves straight in any direction, directedness = 1. CEI is the distance travelled in the direction of the gradient divided by the total distance travelled.

### 2.10. Stimulation of Cells with Folate and cAMP

NC4 or Ax3 cells were grown non-axenically and washed 3 times with KK2 buffer, resuspended at 2 × 10^7^ cells/mL and incubated for 1 h (125 rpm, 22 °C). Cells were stimulated with 200 μM folate and lysed in an LDS sample buffer at 0, 5, 10, 20, 30 45, 60, 90, 120 s and tested by Western blotting.

For cAMP stimulation, 2 × 10^7^ cells/mL cells in KK2 were starved. After 1 h, cells were pulsed with 100 nM cAMP every 6 min for 4 h, and 2 mM of caffeine was added to inhibit internal cAMP signalling. After 1 min of caffeine addition, cells were stimulated with 10 μM of cAMP and lysed in 1X LDS sample buffer at 0, 5, 10, 20, 30 45, 60, 90, 120 s and used for Western blotting.

### 2.11. Microscopy

To determine the morphology, chemotactic speed and directionality of cells phase-contrast time-lapse microscopy were performed at 10x/0.3NA on a Nikon ECLIPSE TE-2000-R inverted microscope equipped with a Retiga EXI CCD monochromatic camera. Images of cells migrating under agarose up a folate gradient were captured every minute for 45 min. DIC images were taken every 2 s for 3 min with 60x/1.4 NA to observe pseudopod formation. HSPC300-eGFP expressing cells were used to determine the activation of the Scar complex. The localization of the Scar/WAVE complex, Arp2/3 complex and F-Actin was examined by a 63x/1.4 NA objective on an AiryScan Zeiss 880 inverted confocal microscope.

The effect of the folate, cAMP, sorbitol or latrunculinA on the recruitment of Scar/WAVE complex was determined by AiryScan confocal microscopy, and *nap1-/egfp-nap1* (eGFPNap1) cells (1 × 10^5^) were seeded in Lab-TekII chambered cover glasses (Thermo Fisher Scientific) and used for imaging after 30 min. Treatments were added gently on the top of cells during imaging. Images were captured every 3 or 30 s.

### 2.12. Quantification of Data and Statistics

Every experiment was performed at least three times. The speed of cells, directedness and CEI were calculated using a homemade plugin in ImageJ. The pseudopodia and Scar/WAVE complex dynamics were calculated by counting the number of frames during a pseudopodia extension manually using ImageJ. Graphs and statistical analyses were performed using Prism.

To quantify the proportion of Abi and Scar/WAVE phosphorylation, the total intensity of all bands and the lowest band were determined using the Odyssey CLx Imaging System (LI-COR Biosciences). The percentage of phosphorylation was calculated by using the formula: % phosphorylation = (Total intensity − intensity of lowest band) * 100/Total intensity.

Statistical significance analyses were performed by non-parametric statistics, such as one-way ANOVA and Dunn’s multiple comparison tests, as described for each result. The experimental groups were tested for normal distribution using the Shapiro-walk test of Prism 7 software (Graphpad, San Diego, CA, USA). Sample sizes are provided in the figure legends. n refers to independently repeated experiments in Western blotting, the total number of cells from 3 independent experiments for cell speed, pseudopod dynamics, Scar patch frequency and duration of patches/accumulation size in ≥25 or more cells.

## 3. Results

### 3.1. Constitutive Phosphorylation of Abi

Analyzing Abi phosphorylation is difficult with standard techniques—normal PAGE lacks resolution, while mass spectrometry poorly detects phosphosites in polyproline domains. In previous studies of Scar/WAVE, we have successfully used low-bis acrylamide SDS-PAGE and Western blotting [13,24]. In normal PAGE gels, Abi from migrating *Dictyostelium* cells runs as a single band (Figure 1A), with quantitative analysis showing this band forming a single diffuse peak (Figure 1B). However, low-bis acrylamide gels resolve identical samples of Abi into two distinct bands (Figure 1C) with clear separation apparent in intensity plots (Figure 1D). The two bands resolve into one upon phosphatase treatment (Figure 1E), and two intensity peaks (solid line; Figure 1F) shift to a single peak (dotted line; Figure 1F), confirming both that the multiple bands are due to phosphorylation and that Abi is phosphorylated in unstimulated cells.

Previous identification of Abi phosphorylation sites relies on prediction, in vitro phosphorylation assays [7,14] or overexpression of tagged proteins [12]. Overexpression of the Scar/WAVE subunits can result in subunits that are not incorporated into a full complex with a strong potential for artifacts. To avoid these secondary effects of overexpression, we explored phosphorylation in native Abi. Mass spectrometry analysis of the full complex, immunoprecipitated using GFP-TRAP on cells with stably tagged GFP-Nap1, did not identify any phosphosites in Abi. However, the proline-rich domain was inefficiently cleaved by proteases—we detected no unphosphorylated peptides in this region following trypsin cleavage, and few (mostly miscleaved) using chymotrypsin—and we have struggled to detect even stoichiometric phosphorylation in the proline-rich region of Scar [13]. We, therefore, screened all the serines in the proline domain (aas 166 to 323) in groups of 1–4 by mutating them into unphosphorylatable alanines (S→A) and testing whether the resulting proteins lost the pattern of bands seen in low-bis Western blots. Simultaneous mutation of three serines—S166,168,169—at the start of the polyproline domain (Abi^S3A^; S166/168/169A) followed by expression in Abi^KO^ cells showed near-total loss of the additional band of Abi (Figure 1G), implying that one or more of these residues is the principal site of phosphorylation. A trace band is just visible, implying a minimal level of phosphorylation at another site. Similarly, phosphomimetic mutation of all three serines to phosphoforms (Abi^S3D^; S166/168/169D)—which is a maximal change, given we could not determine which residues are phosphorylated in vivo—caused a constitutive mobility shift on the gel (Figure 1G). This is also consistent with the additional band of Abi due to phosphorylation. Sequence comparison of *Dictyostelium* Abi with *Homo sapiens* Abi2 in this region indicates conserved phosphorylation sites at serine (S168) and serine (S169) has been substituted by threonine (T) (Appendix A). This suggests that Abi phosphorylation is conserved in *Dictyostelium* and mammals.

### 3.2. Chemoattractant Stimulation of Abi Phosphorylation

Previous studies have shown that extracellular signalling causes cytoskeletal changes and increases protrusions [25,26,27]. This is presumably mediated, in large part, through the Scar/WAVE complex. We have previously shown that phosphorylation of the Scar/WAVE subunit does not activate the complex, nor is it induced by chemoattractant signaling [13], and that the constitutive phosphorylation on the C-terminus of Scar/WAVE makes it less active [28]. In mammalian cells, Abi phosphorylation is thought to be mediated by MAP kinase signalling and by the nonreceptor tyrosine kinase Abl [7,12,14]. *Dictyostelium* lacks nonreceptor tyrosine kinases of the Abl family, and none of the phosphosites of Abi fit the MAP kinase consensus sequence (S/TP). Hence, we examined the changes in the Scar/WAVE redistribution and Scar and Abi phosphorylation in response to chemoattractant stimulus. *Dictyostelium* uses two principal chemoattractants: folate and cAMP, during growth and development, respectively. Treatment of growing cells with folate and developed cells with cAMP causes a rapid change in the amount of polymerized actin [18,29]. Brief exposure of cAMP has also been described to cause rapid recruitment of the Scar/WAVE complex to the cell periphery [30].

To determine the correlation between changes in the Scar/WAVE complex recruitment and phosphorylation in response to chemoattractant stimulus, we examined Scar/WAVE recruitment in cells expressing GFP-Nap1 [28] by AiryScan confocal microscope imaging. Active Scar/WAVE complex accumulates in all-new pseudopods as a short-lived bright patch. In a subset of cells, this response happens in response to folate. More obviously, after folate treatment, the Scar/WAVE, in nearly all cells, redistributes to small puncta at the cell periphery and remains in this state for at least 60 s (insets, Figure 2A, Appendix A). This correlated with a substantial increase in Abi phosphorylation after folate treatment, without affecting Abi expression (Figure 2B,C). In contrast, no significant changes to Scar phosphorylation were observed (Figure 2B,D). In the same way, cAMP stimulation of chemotactically competent cells causes a similar redistribution of Scar/WAVE (insets, Figure 2E) and increased phosphorylation of Abi (Figure 2F,G, Appendix A). A minor but statistically insignificant increase in Scar phosphorylation was also observed in response to cAMP (Figure 2F,H). These results suggest that Abi, but not Scar, phosphorylation connects chemoattractant signalling to the actin cytoskeleton.

To further support this finding, we measured Abi phosphorylation in signalling deficient G-protein beta knockout (Gβ-) cells [31]. As expected, Abi phosphorylation was greatly reduced in Gβ- cells and its expression was significantly increased (Figure 2I,J). Thus, chemotactic signalling stimulates an increase in Abi phosphorylation.

### 3.3. Cell-Substrate Adhesion and Osmotic Shock Alter Abi Phosphorylation

Cell-substrate adhesion is an important regulator of pseudopods and lamellipod dynamics. We recently showed that cell-substrate adhesion increases Scar phosphorylation [13,24]. We, therefore, examined the role of cell-substrate adhesion on Abi phosphorylation, exploiting the ability of *Dictyostelium* to grow in suspension and adhesion. First, we allowed suspended cells to adhere to a Petri dish and followed changes in Abi phosphorylation by Western blotting. This revealed an obvious (and significant—*p* = 0.0082 one-way ANOVA) increase in Abi phosphorylation in adhered cells compared to suspended cells (Figure 3A,B). The phosphorylated band fraction increased from 39.7 ± 12.3% to 66 ± 8.9% (mean ± SD; Figure 3B). In contrast, when we de-adhered cells using a stream of growth medium from a pipette, we observed a clear drop in the relative abundance of the phosphorylated upper band (Figure 3C). The phosphorylated Abi dropped from 50.3 ± 3.9% to 33.8 ± 2.4% of the total (mean ± SD; Figure 3D). Thus, as with Scar phosphorylation, cell-substrate adhesion induces Abi phosphorylation.

*Dictyostelium*, like other chemotactic cells, change shape and reorganize their actin cytoskeleton upon stimulus by chemoattractant. Likewise, cells also cope with the hyper-osmotic stress by re-organizing F-actin and changing their shape [32,33,34]. Additional proteins that translocate to the cell cortex in response to the hyper-osmotic stress are myosin [35], S-adenosyl-L-homocysteine hydrolase (SAHH) and cofilin [32]. A correlation between hyper-osmotic stress, crosslinking of α-actinin and gelation factor with F-actin has been observed earlier [33]. In cell protrusions, F-actin polymerization is principally catalyzed by the Scar/WAVE complex, but the effect of osmotic stress is unknown. We examined live-cell shape changes by DIC microscopy and the recruitment of the Scar/WAVE complex and F-actin distribution by AiryScan confocal imaging after adding sorbitol (0.4 M) to cells attached to a glass surface. Cells retracted their protrusions, stopped movement and shrunk quickly after the addition of sorbitol (Appendix A). The Scar/WAVE complex from the pseudopods (arrow; Appendix A, Panel1) translocated to the cell cortex upon sorbitol treatment (Appendix A). Initially, F-actin localization was mainly in the pseudopod (arrow; Appendix A, panel 1) and redistributed not only to the cell cortex, but also substantially to large vesicles, presumably internalised macropinosomes (asterisks; Appendix A).

Redistribution of the Scar/WAVE complex from the pseudopods to cell cortex suggests changes in its activity (Appendix A) in response to hyper-osmotic stress. We, therefore, examined Abi phosphorylation after cells were treated with various concentrations of sorbitol. Phosphorylation of Abi was unaffected by low concentrations of sorbitol (0.1 M), but was somewhat inhibited by high concentrations (≤0.4 M) (Figure 3E,F). High concentrations of sorbitol (0.5–0.8 M) induced Abi phosphorylation (Figure 3E,F) without affecting total Abi (Figure 3G). This result shows that Abi phosphorylation is affected by multiple changes in the extracellular environment.

### 3.4. Abi Phosphorylation Is Activation-Dependent

The phosphorylation of Scar and Abi is often reported as events upstream of Scar/WAVE complex activation [7]. That is, most authors assert that Scar/WAVE complex members are phosphorylated in order to make them active or to increase the probability that they will be activated. However, we recently found that phosphorylation of Scar is a consequence of the complex activation and not a cause [13]. This has important implications for the mechanism of activation—it implies that phosphorylation’s role is unimportant to the mechanism of complex activation.

To investigate whether Abi phosphorylation is a post-activation event, like Scar/WAVE, we tested the effect of latrunculin A (an inhibitor of actin polymerization [36,37]) on Scar/WAVE recruitment and Abi phosphorylation. Cells treated with latrunculin but no chemoattractant or upstream signal showed exaggerated recruitment of the Scar/WAVE complex and hyperactivation of the Arp2/3 complex (Figure 4A, Appendix A) along with induction of Scar phosphorylation in *Dictyostelium* cells [13,38]. As with Scar [13], Abi phosphorylation increased rapidly upon latrunculin treatment (Figure 4B), with similar kinetics to the increases in Scar and Arp2/3 recruitment. The fraction of Abi that was phosphorylated increased (Figure 4C), with a slight reduction in total Abi upon latrunculin A treatment (Figure 4D). This suggests that Abi phosphorylation is driven by Scar/WAVE activation, rather than upstream signalling.

Scar/WAVE complex activation completely depends on GTP-bound Rac1 interacting with the Pir121 subunit [3,39,40]. We, therefore, examined Abi phosphorylation in cells with the A-site mutation in Pir121, which cannot bind Rac1 [3,4]. This makes the cells excellent for testing whether phosphorylation is upstream or downstream of activation—signalling pathways upstream will be unaffected, whereas activation-dependent processes will be lost. As expected, A-site mutants activated neither the Scar/WAVE nor the Arp2/3 complex (Figure 4E,F), confirming the inactive state of the Scar/WAVE complex. Abi phosphorylation was substantially diminished in the A-site mutant (Figure 4G,H). Furthermore, treatment with latrunculin A, which causes active Scar/WAVE complex to accumulate through an unknown mechanism [3,4], greatly increased the abundance of phosphorylated Abi in normal cells, but only slightly in A-site mutant cells (Figure 4I,J). We note that latrunculin caused wild type Abi to be hyperphosphorylated, producing a higher band that was not seen in normally stimulated cells. It is not clear whether this reflects a second phosphorylation on S166, 168, 169, or new phosphorylation on a different site. Taken together, these results confirm that Abi phosphorylation is a result of Scar/WAVE complex activation.

### 3.5. Abi Phosphorylation Tunes Cell Migration and Pseudopod Formation

In normal cells, every lamellipod’s or pseudopod’s Arp2/3-mediated actin polymerization is driven by the Scar/WAVE complex [2]. There have been several analyses of phosphorylation’s role. Tyrosine phosphorylation in Abi has been implicated in the proper localization of the Scar/WAVE complex in Drosophila [12]. Multiple growth factor-stimulated and Erk-dependent phosphorylation sites (S183, 216, 225, 392, 410 and T265, 267, 394) have been reported to increase Scar/WAVE’s interaction with the Arp2/3 complex [6,14]. However, all these studies were done with either fluorescently tagged, overexpressed or bacterially expressed Abi in vitro, meaning they are at risk of being nonphysiological. To observe the effects of Abi phosphorylation as part of the normal Scar/WAVE complex in moving cells, we co-expressed unlabelled Abi and phospho-mutants (Abi^S3A^ and Abi^S3D^) and HSCP-300-eGFP [41] in Abi null (Abi^KO^) cells. Both phospho-mutants were expressed at the normal level. Western blotting of Pir121, Nap1, Scar and Abi from pull-down samples showed that Abi^WT^, Abi^S3A^ and Abi^S3D^ formed stable complexes with them (Appendix A). To analyze the migratory phenotype, we examined cells migrating under agarose up a folate gradient. As usual, the Scar/WAVE complex localized at the pseudopodia of Abi^WT^, Abi^S3A^ and Abi^S3D^ cells, with marked differences in dynamics (Figure 5A, Appendix A). The Scar/WAVE complex is recruited to the leading edges of cells, in patches (defined here as a coherent area of Scar complex at the leading edge, that lasts 6 s or more [13]) whose size and lifetime anticipate the future pseudopod [13]. The Scar patches of Abi^WT^ recruitment lasted 11.8 ± 4.6 s (mean ± SD), and the phosphorylation-deficient Abi (Abi^S3A^) remained at the pseudopod edges for 12.3 ± 5.3 s (Mean ± SD; Figure 5B). However, the Abi^S3D^ recruitment was short-lived (8.2 ± 2.6 s, mean ± SD; Figure 5B) and Abi^S3D^ patches were smallest at the accumulation site (Figure 5C). This led to changes in the patch frequency. The smaller and shortest-lived Abi^S3D^ patches were made at a higher frequency than Abi^WT^ and Abi^S3A^ (Figure 5D). These results imply that Abi phosphorylation modulates the dynamics of the Scar/WAVE patches.

Cells without Abi migrate poorly [22], mainly using blebs rather than actin pseudopods and are very slow (Figure 5E–G, Appendix A) and less directional (Figure 5G,H). In contrast, Abi^KO^ cells rescued with Abi^WT^, Abi^S3A^ and Abi^S3D^ formed pseudopods that split frequently (Figure 5E, Appendix A). The migration speed and directionality of Abi^KO^ was completely rescued by Abi^WT^ or either phospho-mutant Abi with a decrease in the speed of cells with Abi^S3D^ (Figure 5F–H). Furthermore, pseudopod dynamics were altered in mutants (Figure 5I,J, Appendix A). Abi^S3A^ cell pseudopodia lasted longer (21.1 ± 12 s, mean ± SD) than Abi^WT^ (15.6 ± 7.7 s, mean ± SD), while Abi^S3D^ pseudopods were shorter-lived (11.8 ± 4.9 s, mean ± SD). Moreover, pseudopod generation was more frequent in Abi^S3D^ than in Abi^S3A^ (Figure 5J). We interpret this to mean that the smaller, shorter-lived pseudopods in the phosphomimetic mutant were replaced more frequently. These results clearly show that Abi phosphorylation tunes the dynamics of pseudopods after they are formed.

### 3.6. Double Abi/Scar Phospho-Mutants Still Rescue Migration in Scar-/Abi-Cells

Previously, we have shown that phosphorylation of the Scar/WAVE subunit controls pseudopodia dynamics [13,28]. One hypothesis, combining the data, the results in this paper and the established literature, would be that phosphorylation of either Scar/WAVE or Abi is required for the complex to be activated. When one subunit is mutated, phosphorylation on the other is sufficient to compensate.

We, therefore, tested the combined effects of Scar/WAVE and Abi phosphorylation on the activation of the complex and cell migration. To examine the cells’ physiological behaviour accurately, we generated a complex parent. We created a Scar-/Abi- double knockout and then replaced the endogenous Pir121 with a single copy Pir121-eGFP (Scar-/Abi-/Pir121-eGFP). This yielded a strain in which the Scar/WAVE complex could be followed without tags on either Scar or Abi, or competition with the endogenous proteins, and in which the complex was homogeneously GFP-tagged, giving a high signal/noise ratio in microscopy.

Co-expression of Scar and Abi, as wild type (Scar^WT^/Abi^WT^), unphosphorylatable (Scar^S8A^/Abi^S3A^) and phosphomimetic (Scar^S8D^/Abi^S3D^) mutants, resulted in consistent expression levels of all complex subunits tested, confirming that phosphorylation is not important for complex formation (Figure 6A,B). We examined the different mutants’ migration in detail, using a folate gradient under agarose. Cells expressing wild-type and mutant Scar/Abi formed pseudopodia with altered dynamics (Figure 6C, Appendix A). Scar-/Abi-cells barely form pseudopodia, and those formed are very short-lived, but transfection with all mutants of Scar and Abi rescued pseudopodia formation. Compared with Scar^WT^/Abi^WT^ and Scar^S8D^/Abi^S3D^, Scar^S8A^/Abi^S3A^ formed fewer, more persistent, longer-lived pseudopodia (Figure 6C, Appendix A) Scar^WT^/Abi^WT^. The migration speed was also rescued in the Scar-/Abi- cells by expression of wild type or mutant Scar/Abi (Figure 6D,E). There was a marked increase in the speed of Scar^S8A^/Abi^S3A^ compared with Scar^WT^/Abi^WT^ and Scar^S8D^/Abi^S3D^ (Figure 6D,E). Recruitment of the Scar/WAVE complex was rescued in wild type and cells expressing both pairs of phospho-mutants (Figure 6F, Appendix A), though there were obvious differences. The Scar patches lasted for 11 ± 5.3 s (mean ± SD, *n* = 167) in Scar^WT^/Abi^WT^ (upper panel 1; Figure 6F, Appendix A), while Scar^S8A^/Abi^S3A^ patches’ lifetimes increased to 16 ± 7.6 s (mean ± SD, *n* = 120) (middle panel; Figure 6F, Appendix A up to frame 36 s). In contrast, Scar^S8D^/Abi^S3D^ patches were brief (8.7 ± 3.4 s, mean ± SD, *n* = 141) and highly oscillatory (lower panel; Figure 6F, Appendix A). Overall, these results suggest that neither Scar nor Abi phosphorylation is required for the Scar/WAVE complex activation. Rather, both sets of phosphorylation are tools that enable the regulation of pseudopodia after they are formed. Each phosphorylation on either Scar/WAVE or Abi shortens the lifetime of the Scar/WAVE complex patches and, therefore, the distance covered by each pseudopod.

## 4. Discussion

Abi phosphorylation is seen in all species examined. It occurs in the same general region—near the polyproline domain—but the exact sites are not conserved. This is hardly surprising. In both Scar/WAVE and Abi, the polyproline domains are very variable; their existence is conserved between species, but their sequence is not. This raises interesting questions about Abi phosphorylation’s mechanisms and biological functions. It is presumably not creating a simple binding site, or it would be conserved in more detail. Our work in this paper shows that it only occurs after the Scar/WAVE complex has been activated, a process that is thought to involve a tightly folded, autoinhibited complex being converted into an open array with multiple binding sites for other proteins, in particular, effectors of actin polymerization like the Arp2/3 complex and VASP. Scar/WAVE itself appears to show similar behaviour; it is phosphorylated only after the complex has been recruited to pseudopods. Thus it is unlikely that phosphorylation of either Scar/WAVE or Abi is fulfilling the role supported by many earlier papers, of causing or allowing the complex to be activated. This was clearly shown by our double mutants—Scar/WAVE complex in which neither Abi nor Scar/WAVE itself can be phosphorylated is still active. Rather, it is likely that phosphorylation modulates the stability of the open complex or modulates its affinity for its binding partners.

We have not identified the precise sites of Abi phosphorylation. We predominantly see only two bands on low-bis gels, one corresponding to the unphosphorylated form and one other. By analogy with Scar, where we see multiple bands, there may typically only be one phosphate per Abi molecule located on a serine (S166, S168 or 169). The latrunculin experiment (Figure 4) shows that more phosphorylations (at least three per molecule) may occur in extremis; the phosphorylation site is clearly not unique. This is strongly analogous to the results of studies on the cAMP receptor cAR1. The wild type phosphorylation site was identified, but mutating it did not cause discernible reduction in phosphorylation. Instead, all the serines in the vicinity of the site had to be mutated to abolish the phosphorylation [42]. None of the serines has an obvious consensus site, making it unlikely that signaling kinases like ERKs or PKB are involved. This is consistent with phosphorylation occurring after activation, rather than causing it. It is, however, entirely clear that removing essentially all the phosphorylation does not affect the Abi’s ability to be activated, so more detailed mutagenesis might more precisely localize the phosphosite but would probably not add to the biological insight.

The precise mechanism by which Abi phosphorylation modulates the Scar/WAVE complex is unclear. This is because so much is yet to be resolved about the complex’s regulation. It is clear that Rac plays a key role, but there are many other interactors that are thought to be important and whose mechanisms of action are not known. In particular, the Scar/WAVE complex patches we observe are maintained by positive feedback loops. Phosphorylation could slightly diminish the strength of the positive feedback. A small change in this would effect a large change in the lifetime of the assemblies. An alternative explanation could involve protein stability. The Scar/WAVE complex is regulated by proteolysis in a complicated way—loss of either of the largest subunits (PIR121 and Nap1) causes a catastrophic loss of Abi, Scar/WAVE itself and HSPC300/Brk1, but loss of Abi does not cause loss of the PIR121/Nap1 dimer. We interpret this to mean that there are at least two proteolytic pathways, one that regulates intact complex and PIR121/Nap1, and a far more active one that is specific to the smaller subunits. Phosphorylation of Abi could increase the rate of either pathway or make the activated complex shorter-lived. It is particularly interesting that Abi phosphorylation, unlike Scar, seems to be regulated by chemoattractants. This implies that the changes in stability may affect the lifetime of pseudopods, which can be a key regulator of chemotaxis [43,44]. As all our mutant cells chemotax efficiently, Abi phosphorylation is plainly not an essential player.

Overall, this work does not lessen the importance of Abi phosphorylation. It may not be a key activating step, but we have shown that it changes the extent to which the Scar/WAVE complex promotes actin polymerization and thus modulates the cell’s movement speed. In an essential process like cell motility, this is a vital role.

## Figures and Tables

**Figure 1 cells-10-03485-f001:**
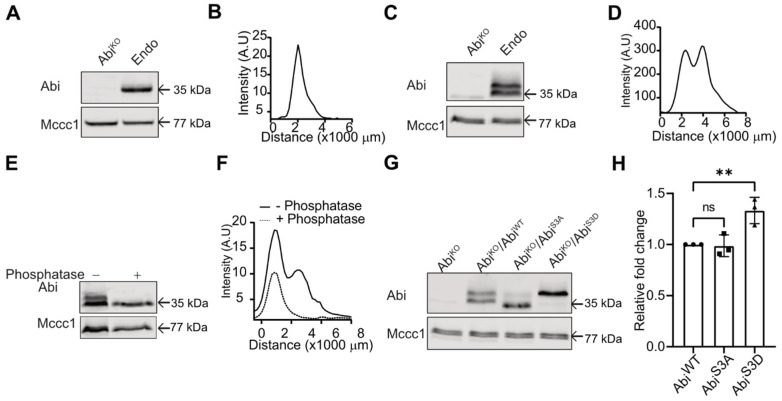
Multiple phosphorylations in Abi. (**A**) Western blot of *Dictyostelium* Abi using normal (0.3%) bis−acrylamide gels. Whole cells were lysed in sample buffer, boiled and separated on 10% bis−tris SDS-PAGE gels, blotted, and probed with an anti-Abi antibody, yielding a single band. Mccc1 is a loading control. (**B**) Line scan of Abi band signal intensity from normal gels shows a single peak. (**C**) Western blot of Abi using low (0.06%) bis-acrylamide gels, yielding two discrete bands. (**D**) Line scan of Abi bands’ signal intensities from low-bis gels shows clear separate peaks. (**E**,**F**) Phosphatase treatment. Whole *Dictyostelium* cells were lysed in TN/T buffer, then incubated with and without lambda phosphatase, boiled in sample buffer and analyzed using low-bis acrylamide gels. The scan shows two peaks resolved into one and shifted downwards after phosphatase treatment. (**G**) Unphosphorylatable (Abi^S3A^) and phosphomimetic triple (Abi^S3D^) mutations of serines S166/168/169 (S3) analysed by Western blotting on low-bis gels. Unphosphorylatable mutants run as a single, high-mobility band; phosphomimetic ones run as a single low-mobility band. These experiments were repeated three times. (**H**) Quantification of Abi expression in Abi^WT^, Abi^S3A^ and Abi^S3D^ cells. Abi expression was slightly increased in Abi^S3D^ cells but was similar in Abi^WT^ and Abi^S3A^ cells. (*n* = 3, mean ± SD, ns = not significant, ** *p* = 0.0098, Dunnett’s multiple comparison test). KO; knock-out, Endo; endogenous.

**Figure 2 cells-10-03485-f002:**
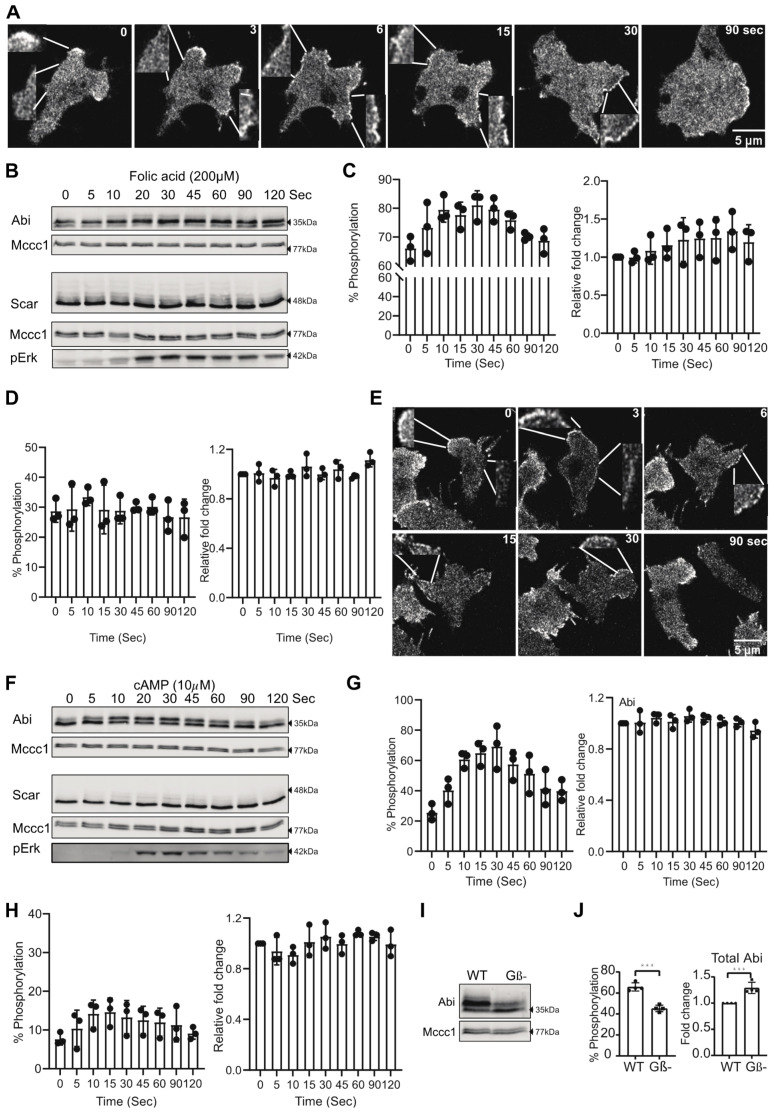
Signaling dependent phosphorylation of Abi and Scar. (**A**) Redistribution of the Scar/WAVE complex after folate treatment. *Dictyostelium* cells expressing EGFP-Nap1, which labels the Scar/WAVE complex, were imaged by AiryScan confocal microscopy. The Scar/WAVE complex rapidly gets redistributed as puncta on the cell membrane periphery after addition of folate (200 µM) treatment (Scale = 5 µm). (**B**–**D**) The effect of folate stimulation on Abi and Scar phosphorylation. Growing *Dictyostelium* cells were washed and treated with 200 µM folate for the indicated time points, then Abi and Scar band shifts were analyzed using low Bis-acrylamyde gels. Folate enhances Abi phosphorylation (**C**; mean ± SD, *n* = 3, *p* = 0.037, Kruskal Wallis test) but not Scar phosphorylation (**D**; mean ± SD, *n* = 3, *p* = 0.8, Kruskal Wallis test) despite stimulation of signaling shown by Erk phosphorylation. Total Abi and Scar expression remains unaffected. (**E**) Redistribution of the Scar/WAVE complex after cAMP treatment. cAMP competent *Dictyostelium* cells expressing EGFP-Nap1 were imaged by AiryScan confocal microscopy. The Scar/WAVE complex gets redistributed rapidly on the cell membrane periphery after addition of cAMP (10 µM) treatment (Scale = 5 µm). (**F**–**H**) The effect of cAMP stimulation on Abi and Scar phosphorylation. cAMP-competent cells were stimulated with 10µM cAMP for the indicated time points, then Abi and Scar band shifts were analyzed by Western blotting. cAMP enhances Abi phosphorylation (**G**; mean ± SD, *n* = 3, *p* = 0.014, Kruskal Wallis test) substantially, with little change in Scar phosphorylation (**H**; mean ± SD, *n* = 3, *p* = 0.3, Kruskal Wallis test) despite huge activation of signalling shown by Erk phosphorylation. (**I**,**J**) Abi phosphorylation in WT and Gβ- cells. Abi band shifts were analyzed in WT and Gβ- cell by Western blotting. Abi in Gβ- cells has substantially less phosphorylation and more expression (**J**; mean ± SD; *n* = 4, *** *p* < 0.001, unpaired *t*-test). cAMP, cyclic 3′,5′-adenosine monophosphate; ERK, extracellular signal-regulated kinase; Gβ, G-protein beta subunit; WT, wild type.

**Figure 3 cells-10-03485-f003:**
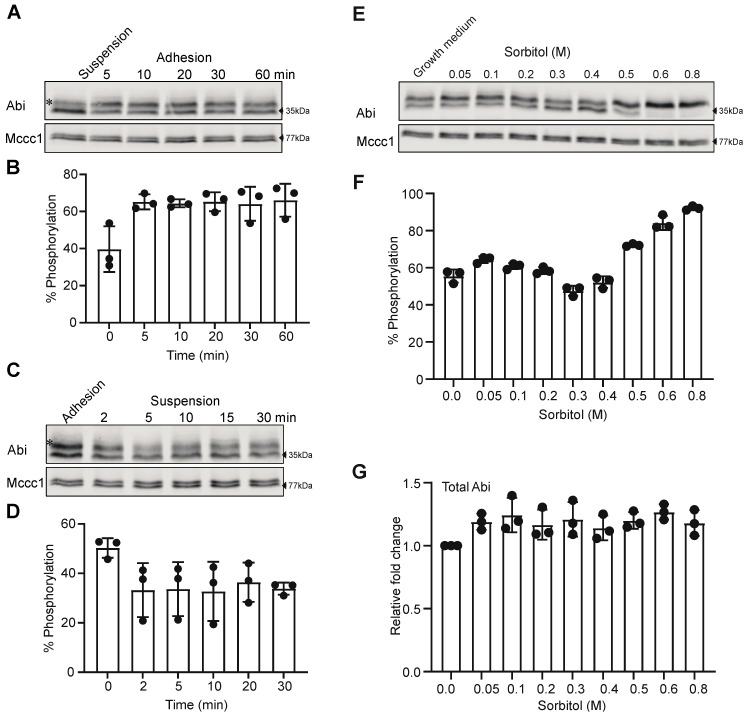
Cell-substrate adhesion enhances Abi phosphorylation. (**A**,**B**) Effect of adhesion on Abi phosphorylation. Suspension grown cells were allowed to adhere in Petri dishes and then lysed at indicated time points, and cell lysates were analyzed by Western blotting. The intensity of the upper Abi band (*) increases after adhesion. Quantitation of lane band intensities (intensity of “upper bands”/total band intensity; mean ± SD, *n* = 3, *p* = 0.0082, Ordinary one-way ANOVA) confirms increased intensities of phosphorylated Abi bands after adhesion (**B**). (**C**,**D**) Effect of de-adhesion on Abi phosphorylation. Cells grown in adhesion on Petri dishes were detached by a jet of medium from a 10 mL pipette, collected and maintained in suspension (approx. 2 × 10^6^ cells/mL) by shaking at 120 rpm. Cell lysates were analyzed by Western blotting. Quantitation of lane intensities (mean ± SD, *n* = 3, *p* = 0.19, Ordinary one-way ANOVA) confirms a loss of phosphorylated Abi bands (*) intensity (**D**). (**E**–**G**) Effect of various sorbitol concentrations on Abi phosphorylation. Medium from cells was replaced with PB containing various concentrations of sorbitol. After 5 min cell lysates were prepared in sample buffer and analyzed for Abi (**E**) band shifts. Abi phosphorylation is abolished at 400 mM sorbitol and induced with >400 mM sorbitol (**F**; mean ± SD, *n* = 3, *p* = 0.0001, Ordinary one-way ANOVA) without affecting total Abi concentration (**G**; mean ± SD, *n* = 3, *p* = 0.3, Ordinary one-way ANOVA).

**Figure 4 cells-10-03485-f004:**
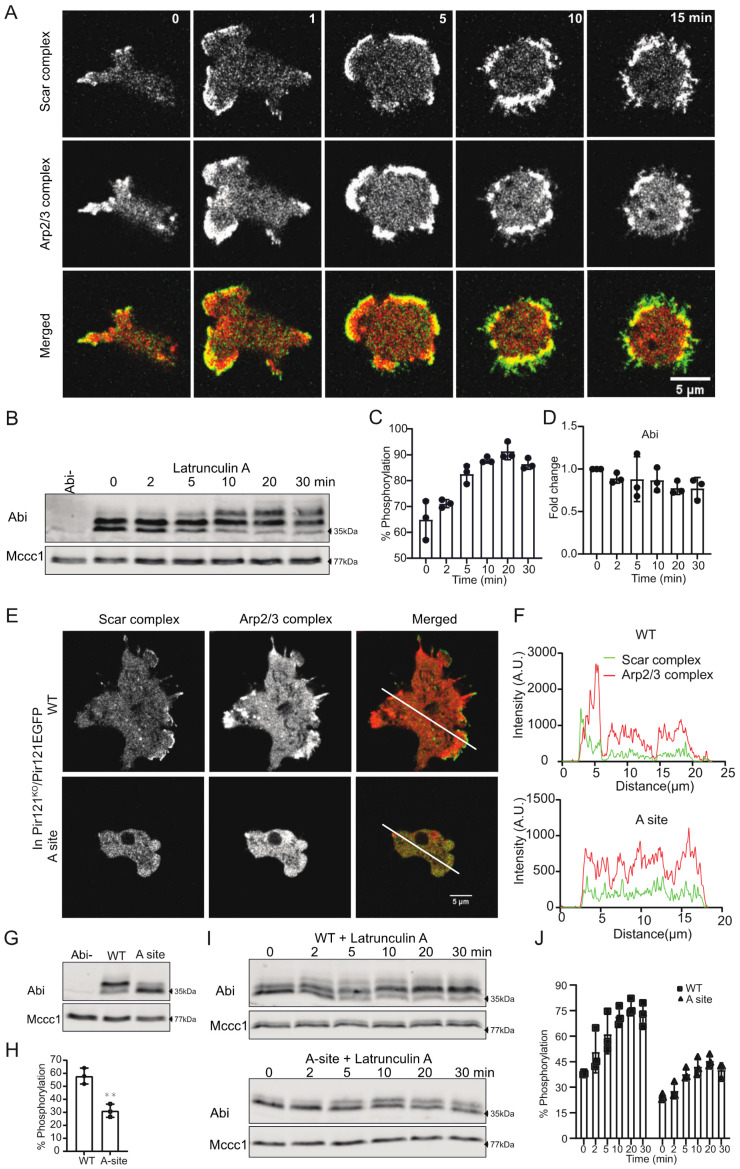
Effect of the activation of the Scar/WAVE complex on Abi phosphorylation. (**A**) Hyperactivation and redistribution of the Scar/WAVE complex after latrunculin A treatment. Cells expressing EGFP-Nap1 and mRFP-mars2-ArpC4 were imaged using an AiryScan confocal microscope. Latrunculin A (5 μM) was added at t = 30 s. Scale bar = 5 μm. Addition of latrunculin enhances the Scar complex and Arp2/3 recruitment at the cell periphery. This experiment was repeated 3 times. (**B**–**D**) Increased Abi phosphorylation after latrunculin A treatment. Cells treated with latrunculin A for the indicated times were lysed in sample buffer, then Abi band shift and intensities were analyzed by Western blotting. Phosphorylated Abi bands become more abundant due to latrunculin A treatment (**C**; mean ± SD, *n* = 3, *p* = 0.0001, Ordinary one-way ANOVA), with a slight reduction in total Abi (**D**; mean ± SD, *n* = 3, *p* = 0.4, Ordinary one-way ANOVA). (**E**) Importance of A-site in the activation of the Scar/WAVE complex and Arp2/3 complex. WT and the A-site mutant of Pir121 were co-expressed with mRFP-mars2-ArpC4 in the Pir121- cells and imaged by AiryScan confocal microscopy under agarose chemotaxis up a folate gradient. The A-site mutant is unable to activate the Scar/WAVE and Arp2/3 complexes. This experiment was repeated 3 times. (**F**) Quantitation of Scar and Arp2/3 activation in WT and A-site mutant from merged panels. A line was drawn across the centre of the cell and relative pixel values plotted. (**G**,**H**) Importance of Rac1-binding PIR121 A site for Abi phosphorylation. WT, A site (K193D/R194D) Pir121 expressing cells were analyzed for Abi band shifts by Western blotting. Multiple phosphorylated Abi bands are absent in A site mutant. Abi phosphorylation is significantly reduced in A site cells (**H**; mean ± SD, *n* = 3, ** *p* = 0.004, unpaired *t*-test). (**I**,**J**) Effect of latrunculin A treatment on Abi phosphorylation in WT and A-site mutant. WT and A site mutant of Pir121 were treated with latrunculin A for the indicated times, lysed in sample buffer, then Abi band shift and intensities were analyzed by Western blotting. Phosphorylated Abi bands become abundant in WT cells after latrunculin A treatment. Quantification of Abi phosphorylation shows substantial increase in Abi phosphorylation in WT cells compared with A site (**J**; mean ± SD, *n* = 3, *p* = 0.0046, unpaired *t*-test). A.U.; arbitrary unit, WT; wild type, A- site; adjacent site.

**Figure 5 cells-10-03485-f005:**
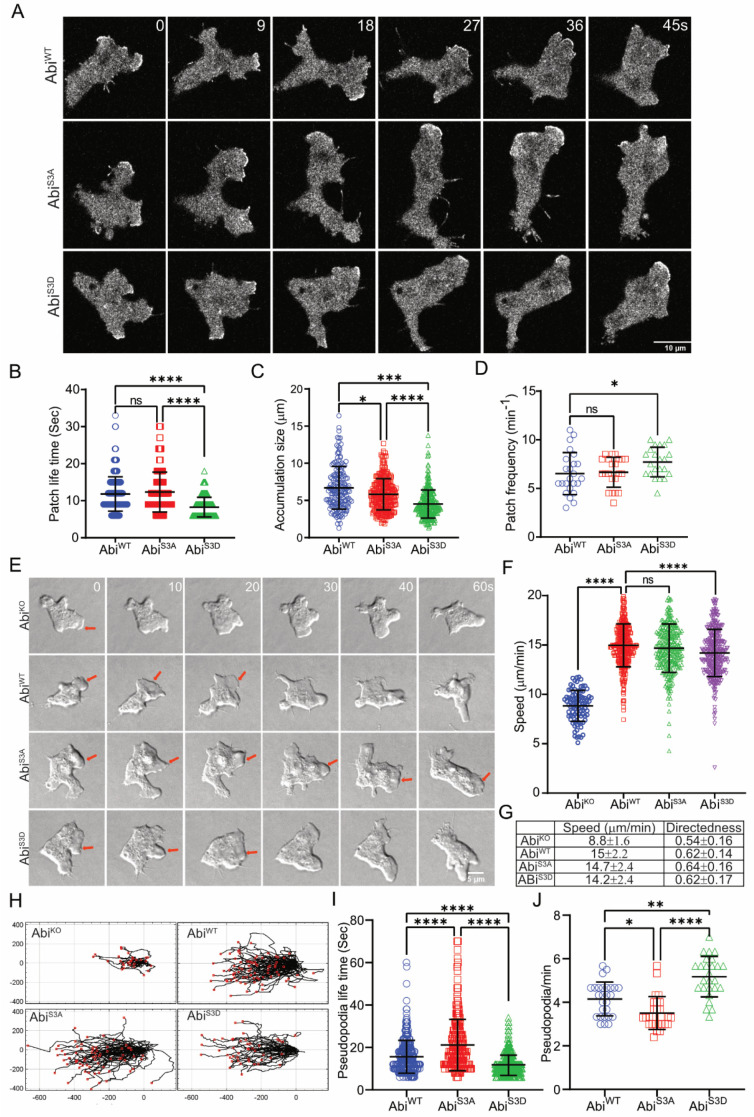
Abi mutants are recruited to pseudopodia and are functional. (**A**) Subcellular localization of Scar complex. Abi^WT^, Abi^S3A^, and Abi^S3D^ were co-expressed with HSPC300−eGFP in Abi- cells, allowed to migrate under agarose up a folate gradient and examined by AiryScan confocal microscopy at a frame interval of 3 s (1f/3s). All show efficient Scar/WAVE complex localization to patches at the pseudopods. (**B**–**D**) Lifetimes, size, and generation rates of mutant Scar patches. Patch lifetime was measured from the number of frames a patch showed the continuous presence of labelled Scar complex, size was measured as the peak length of contiguous recruitment along the circumference of the cell from individual frames and generation frequency was calculated from the number of patches lasting at least 2 frames. Abi^WT^ and Abi^S3A^ patches are less frequent and long−lived compared to Abi^S3D^ (mean ± SD; *n* > 25 cells over 3 experiments; ns = not significant, * *p* ≤ 0.05, *** *p* ≤ 0.001, **** *p* ≤ 0.0001, 1-way ANOVA, Dunn’s multiple comparison test). (**E**) Rescue of pseudopod formation by mutated Abi. Abi-cells were transfected with Abi^WT^, Abi^S3A^, and Abi^S3D^ and allowed to migrate under agarose up a folate gradient while being observed by DIC microscopy at a frame interval of 2 s (1f/2s). All rescued cells formed actin pseudopods; those formed by Abi^S3A^-expressing cells were longer lived and Abi^S3D^-expressing cells were short-lived compared to Abi^WT^. Arrows indicate blebbing in Abi^KO^ and a single pseudopod in rescued cells. Panel (**F**) shows cell migration speed (mean ± SD; *n* = 90 Abi-, 327 WT, 296S3A, 347S3D over 3 independent experiments, 1-way ANOVA, Dunn’s multiple comparison test). (**G**) Trajectories of cells migrating under agarose up a folate gradient. (**H**) Summary of chemotaxis features of WT and phosphomutants. (**I**,**J**) Abi^WT^, Abi^S3A^ and Abi^S3D^ on pseudopod dynamics. Pseudopod lifetime and frequency were measured by counting the number of pseudopods lasting at least 2 frames from DIC videos. Abi^S3A^ yields increased, and Abi^S3D^ demonstrated decreased lifetimes. Abi^S3A^, pseudopods generated less frequently than Abi^WT^ and Abi^S3D^, while Abi^S3D^ generated pseudopods which were more frequent than Abi^WT^ and Abi^S3A^ (mean ± SD; *n* > 25 cells over 3 experiments; * *p* ≤ 0.05, ** *p* ≤ 0.01, **** *p* ≤ 0.0001, 1-way ANOVA, Dunn’s multiple comparison test).

**Figure 6 cells-10-03485-f006:**
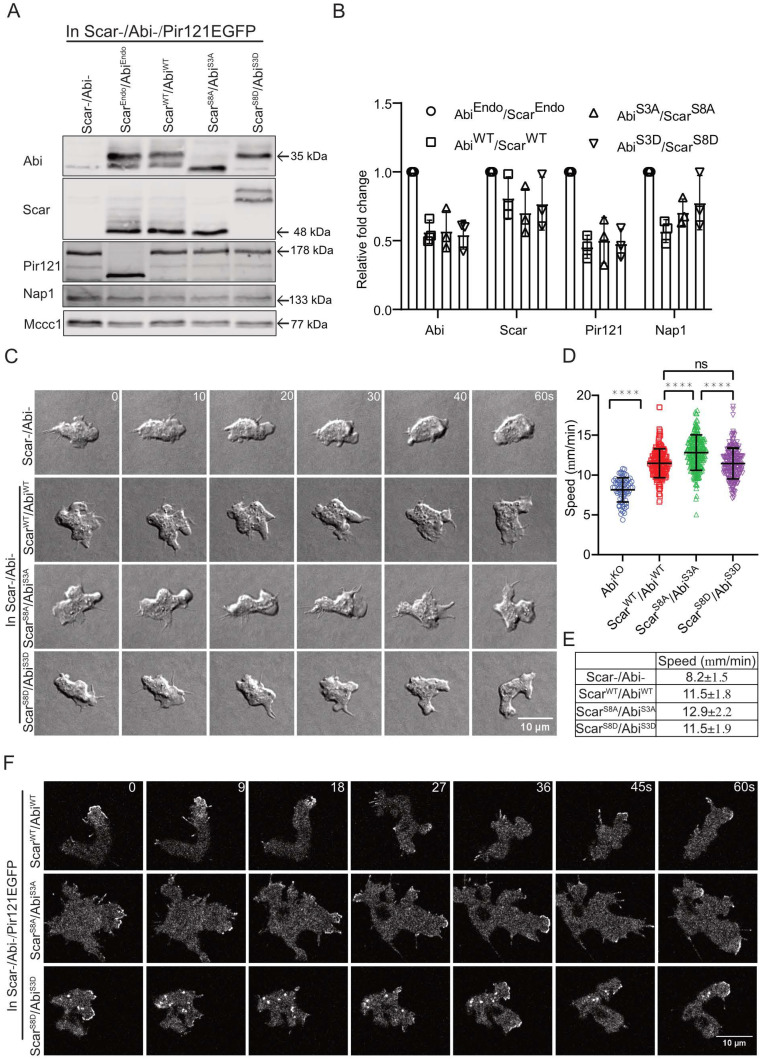
Both Scar and Abi phospho-mutants are recruited to pseudopodia and are functional. (**A**) Stability of the WT and phosphomutant Scar/WAVE complex. Scar^WT^/Abi^WT^, Scar^S8A^/Abi^S3A^ and Scar^S8D^/Abi^S3D^ were expressed in Scar-/Abi-/Pir121-eGFP cells. Expression of Abi, Scar, Pir121 and Nap1 was compared by Western blotting. WT and mutant Scar/Abi expression restored the expression of all subunits. Mccc1 was used as a loading control. (**B**) Densitometric quantification of protein bands. (**C**) Rescue of pseudopod formation by WT and phosphomutants of Scar/Abi. Scar-/Abi-/Pir121-eGFP cells were transfected with Scar^WT^/Abi^WT^, Scar^S8A^/Abi^S3A^ and Scar^S8D^/Abi^S3D^ and allowed to migrate under agarose up a folate gradient while being observed by DIC microscopy at a frame interval of 2 s (1f/2s). (**D**) Shows speeds (mean ± SD; *n* = 65 Scar-/Abi-, 193 Scar^WT^/Abi^WT^, 233 Scar^S8A^/Abi^S3A^ and 195 Scar^S8D^/Abi^S3D^ over 3 independent experiments, **** *p* ≤ 0.0001, 1-way ANOVA, Dunn’s multiple comparison test). (**E**) Summary of chemotaxis features of WT and phosphomutants. (**F**) Subcellular localization of the Scar complex. Scar^WT^/Abi^WT^, Scar^S8A^/Abi^S3A^ and Scar^S8D^/Abi^S3D^ were expressed in Scar-/Abi-/Pir121-eGFP cells, which were allowed to migrate under agarose up a folate gradient and were examined by AiryScan confocal microscopy at a frame interval of 3 s (1f/3s).

## Data Availability

All the raw data for the figures and videos used in this study can be requested from the corresponding author (Robert.Insall@glasgow.ac.uk) and lead author (s.singh@beatson.gla.ac.uk).

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
