# Peer review of "Extracellular Signalling Modulates Scar/WAVE Complex Activity through Abi Phosphorylation"

_cells, 2021, doi:10.3390/cells10123485_

Round 1

Reviewer 1 Report

In this manuscript, Singh et al. explore the regulation of the Scar/WAVE complex through phosphorylation of one of its subunits Abi in the model organism Dictyostelium. They report that Abi is phosphorylated in response to cell adhesion and treatment with chemoattractants. They define the Abi Serine phosphorylation sites by site directed mutagenesis without revealing how they identified these sites. This study follows a similar one in which the authors explored the phosphorylation of Scar itself. The authors claim that Abi is only phosphorylated after the Scar/WAVE complex is activated which they define as binding to active Rac. The authors then explore the functional consequences of Abi phosphorylation using a knockout rescue system in Dictyostelium and report that pseudopodia with unphosphorylatable Abi (or Abi and Scar) protrude for a longer time and the cells migrate faster.

This is a very important, timely research question, as we do not fully understand how the Scar/WAVE complex is controlled in time and space.

Major comments:

  1. The biggest issue I have with this manuscript is that it makes broad generalised claims about the role of phosphorylation of Abi for Scar/WAVE complex activation from studying the complex in Dictyostelium, which is a good model organism but as the authors themselves note do not have the receptor tyrosine kinases which are found in mammalian cells. The authors acknowledge this fact but then ignore this when discussing their results and making broad, generalised statements including in the title. Either the authors experimentally compare this to the effect of Abi phosphorylation on Scar/WAVE activity in mammalian cells or they have to very carefully describe their results and state that their results may only apply to
  2. Unfortunately, the most important claim is not backed by data: That Abi is only phosphorylated after the Scar/WAVE complex is activated which they define as binding to active Rac. In Fig 4 G,H the upshifted, phosphorylated band of Abi can be clearly seen in the blots for the PIR121 Rac binding defective mutant expressing cells. These blots were also not quantified.
  3. The authors do not discuss their results appropriately in the context of prior knowledge– the authors appear to lightly dismiss all the prior data as artefacts due to overexpression. In a seminal study, Lebensohn and Kirschner, Mol Cell 2009 showed that 3 coincident signals are needed, active Rac, phosphorylation and PIP3 liposomes to activate the native, purified complex. In that study neither overexpression nor protein tagging was used. Of course, I appreciate that in the context of living cells this could be different and it important to test this thoroughly.
  4. Unfortunately, I am not convinced that the authors of this manuscript provide good evidence that in general Abi phosphorylation is not required for Scar/WAVE complex activation as several experiments are not well performed and many are not well described and important information and explanations are missing. In addition, many figure legends are incomplete or confused.

Specific comments:

Statistical analysis is needed for all quantifications of phosphorylations. Why was this only done for Fig 3AB?

Please show molecular weight bands for all blots.

Fig 1: Please indicate in the figure legends how often the experiments were repeated. Please quantify total Abi levels compared to housekeeping gene to allow estimation of total Abi levels in the rescued cell lines.

“Given the clear phosphorylation seen in Fig. 1 this could be due to two reasons - either the phosphosites were inaccessible to trypsin or chymotrypsin cleavage, or they could not be detected by mass spectrometry (a particular problem for phosphosites in the polyproline domain).”

Clarify whether you had the coverage in the mass spec for the proline rich region – you should know whether trypsin or chymotrypsin cleavage worked.

Please add information how you identified the “plausible” sites: “We therefore screened the plausible serines (S) and threonines (T) in Abi” this is important information – was this done by bioinformatics or prior knowledge based on the studies with overexpressed proteins?

“Simultaneous mutation of all three residues to unphosphorylatable (AbiS3A; S166/168/169A) forms and expression in AbiKO cells showed near-total loss of the additional band of Abi (Figure 1G)”

From this data you cannot conclude that all three serines are phosphorylated. Clearly state this lack of knowledge or make single and double mutant combinations to identify the phosphorylated sites.

Fig 1 G: in the S3A mutant the band is lower than the lower band in the Abi WT. Also, in Fig 2B you have 3 bands: at 0 sec a double band and then at 10 sec there is a new higher band appearing.  Again three bands are visible in Fig. 4B and H. This should be described and the bands separately quantified. Which band was quantified in Fig 2C, Fig. 4B?

Fig 2 BFI: Please quantify total Abi levels compared to housekeeping gene to test whether Abi expression is affected by phosphorylation.

Movie S1 and S2: Please provide more details of the Airyscan imaging. Movie S2 has several frames that “jump around” as if someone had kicked the microscope. Please replace it with a better movie.

At a frame every 3 seconds, your sampling rate is not high enough to capture patches (flashes?) and leading edge dynamics.

Fig 2A+E: “after folate treatment, the Scar/WAVE in nearly all cells redistributes to small puncta at the cell periphery, and remains in this state for at least 60 seconds”

It is not clear where in the images are supposed to be puncta– please add arrow heads and make inset to clearly show this.

Fig. 3 “(E-G) Effect of various sorbitol concentrations on Abi and Scar phosphorylation” Where in this figure is Scar phosphorylation shown?

Fig 4A,E: How often was this experiment repeated?

Fig 4B: Again 3 bands are visible; 4C – which band was quantified here?

Fig 4B-D Please normalise to housekeeping gene for levels – it appears that there is a drop in Abi levels.

Fig 4E,F: Please show where the linescan is located.

Fig 4 G/H. How often was this experiment repeated.  Quantification is essential as a major claim of this paper is based on this.

“As expected, A-site mutants activated neither the Scar/WAVE nor the Arp2/3 complex (Figure 4 E&F), confirming the inactive state of the Scar/WAVE complex. Abi phosphorylation was abolished in the A-site mutant (Figure 4G). Furthermore, treatment with latrunculin A did not stimulate Abi phosphorylation in A-site mutant cells (Figure 4H). Altogether, these results confirm that Abi phosphorylation is a result of Scar/WAVE complex activation.”

This claim is not backed by the data shown:

Fig 4H There is a clearly visible increase in phosphorylation on A site mutant for Abi  at 5-30 min – so it does have an effect – please quantify.Show where linescan is.

Figure 5 is not well described:

Fig 5 A: “AbiWT, AbiS3A and AbiS3D all localized at the pseudopodia of cells,”

This claim is not backed by data as Abi proteins were untagged.

Fig 5 A Please define patches. Are you defining this as substructures of the leading edge?

Movie S7: it appears that some of the cells may have been “re-centred” because they moved out of the frame? This is highly unusual and should be avoided as it misrepresents the movie.

Fig 5 B -D Because your frame rate is too low for the fast movement your patch lifetimes appear as “categories” in your quantification. This is problematic as you will miss lots of events due to undersampling.

Please define: “accumulation size”. Is accumulation size an entire leading edge? Is this average data from cells or are individual patches displayed?

5G How was “directness” quantified?

5I Pseudopodia lifetimes appear not equally distributed – please check this and use the correct statistical test.

Fig 6A This figure shows that you only rescue 50% of the levels of endogenous Abi in your rescues – this should be taken into account in your interpretation.

Fig 6D Since you are just rescuing 50% of endogenous levels it is essential to compare the speed measurements also to endogenous and not just KO to rescues.

Minor comments:

Line 102 typo: 1 mg of lysate à 1 mg meant

S2: in C please add label “pulldown” and fix figure legend in which use twice (A)

Reviewer 2 Report

The manuscript by Singh and coauthors explores Ser/Thr phosphorylation of Abi (a component of the SCAR/WAVE complex) and the role of this phosphorylation in Dictyostelium pseudopod formation and cell migration. The authors convincingly demonstrate that the phosphorylation is not essential for activation of the complex but contributes to the regulation of the stability/dynamics of pseudopods. The manuscript is well written, the study is conducted at a good technical level and overall is sound and credible. There are several issues, however, that should be addressed.

Major comments

  1. Neither the abstract nor the title of the work mentioned that the work is conducted on Dictyostelium. Since the authors did not explore other systems (e.g., mammalian cells), such generalization is not justified and should be avoided; the model organism must be indicated at least in the abstract if not in the title.
  2. Statistical methods are described, but the significance is not indicated on the provided graphs.
  3. The authors identified S166/S168/S169 as potential phosphorylation sites in Abi. It is not clear why there was no attempt to identify which of the residues or all three are phosphorylated. Doing that for one or two key experiments would strengthen the study.
  4. On a similar matter, Fig 4B (Abi phosphorylation under LatA), an additional band of Abi accumulates at longer incubation times, likely pointing to more than a single phosphorylation event. Such extra bands do not appear under other tested conditions. Why does Lat A cause activation of WAVE/SCAR and phosphorylation of Abi? Is that phosphorylation in the 166-169 area or a different region? Even if the mechanisms of this phenomenon are not known, they should not be ignored.
  5. Fig 4G,H. The legend states that “phosphorylated Abi bands become abundant in WT cells” and the manuscript body says, “treatment with latrunculin A did not stimulate Abi phosphorylation in A-site mutant cells (Figure 4H)”. Yet, a single extra band of Abi is clearly accumulated under these conditions. Therefore, the description does not seem to match the observation. It is also not clear why the experimental results of Fig4 G,H are not quantified.

Minor comments:

  1. Providing sequence comparison of mammalian and dicty Abi, either in full or near the phosphorylation sites, would be beneficial to appreciate the potential impact of the findings on other organisms.
  2. Lanes 198-199: “This further confirms that the additional band of Abi is due to phosphorylation.”

The chemical nature of the phosphate group is different from that of the aspartate side chain; the number of phosphorylation sites (and therefore, charges) is not established. Therefore the observation is consistent with but does not confirm that the shift results from phosphorylation.

  1. Lanes 190 -194: The transition between these sentences is not clear to me. What is exactly the reasoning behind focusing on these particular residues as potential phosphorylation sites? What are the kinases that can phosphorylate these residues based on the consensus motifs?
  2. Figure 3D: the values within triplicates are exceptionally similar. Was the experiment biological or technical triplicate?
  3. Lane 445: in the phrase “…examine the cell’s behavior accurately and completely…” the latter is unrealistic.
  4. Lane 501-2: What is meant by the following statement? “Scar/WAVE itself is phosphorylated in exactly the same way”.

Round 2

Reviewer 2 Report

Overall, I am satisfied with how my concerns were addressed.

Minor comments:

  1. Fig. 1 has two nonidentical legends. Which one is correct?
  2. Unify capitalization for ANOVA.
  3. Fig. 1D X-axis legend must be clarified.

Author Response

  1.  I think I've understood this.  Panels B and D had the same legend.  I've clarified.

2.  ANOVA has been capitalized throughout.

3.  The new version of 1D should be clearer

We also found an error in Fig. 6 that has now been corrected.